# Physicians’ Disease Severity Ratings are Non-Inferior to the Emergency Severity Index

**DOI:** 10.3390/jcm9030762

**Published:** 2020-03-11

**Authors:** Roland Bingisser, Severin Manuel Baerlocher, Tobias Kuster, Ricardo Nieves Ortega, Christian H. Nickel

**Affiliations:** Emergency Department, University Hospital, CH-4031 Basel, Switzerland; severin.baerlocher@usb.ch (S.M.B.); tobias.kuster@usb.ch (T.K.); ricardo.nieves@usb.ch (R.N.O.); Christian.Nickel@usb.ch (C.H.N.)

**Keywords:** triage, emergency severity index, prospective study

## Abstract

Our objective was to compare informal physicians’ disease severity ratings (PDSR) at presentation with the well-established Emergency Severity Index (ESI) in order to test for non-inferiority of the discriminatory ability regarding hospitalization, intensive care, and mortality. We made a prospective observational study with consecutive enrollment. At presentation, the PDSR and subsequently Emergency Severity Index (ESI) levels were recorded. The primary outcome was the non-inferiority of the discriminatory ability (PDSR versus ESI) regarding hospitalization, intensive care, and mortality. The secondary outcomes were the reliability, the predictive validity, and the safety of PDSR. We included 6859 patients. The median age was 51 years (interquartile range (IQR) = 33 to 72 years); 51.4% were males. There were 159 non-survivors (2.4%) at the 30 day follow-up. The PDSR’s discriminatory ability was non-inferior to the ESI’s discriminatory ability. The safety assessment showed mortality of <0.5% in low-acuity patients in both tools. The predictive validity increased by 3.5 to 7 times if adding high-acuity PDSR to ESI in all categories with mortality of >1%. Our data showed the non-inferiority of PDSR compared with ESI regarding discriminatory ability, a moderate reliability, and an acceptable safety of both tools.

## 1. Introduction

### 1.1. Background

The standardized use of formal triage is often unable to identify high-acuity patients within the recommended time-frame [1]. During times of crowding, waiting for triage hampers the timely identification of time-critical conditions, which is associated with adverse outcomes [2]. Crowding was shown to occur in up to 50% of all shifts in larger emergency departments (EDs) [3]. Therefore, crowding may have detrimental effects—e.g., on recommended door-to-balloon time in patients with ST-elevated myocardial infarction (STEMI) [4]. Possible solutions may be an increase of staff at triage, medical team evaluation [5], or accelerating the triage process.

Informal risk stratification tools have been shown to predict hospitalization and mortality [6,7,8,9]. They have previously been described as clinical intuition [7], informal triage [10], or physicians’ disease severity ratings (PDSR) [8,11]. To date, it is not clear whether and how informal risk stratification can be used in clinical routine. Informally and formally structured triage systems appear to have similar validities [6], and patients at increased risk of dying can be identified informally with acceptable results [7]. However, to ensure transparency and uniformity, a verifiable, formally structured triage system for ED patients is advocated for by most authors. Nevertheless, in times of crowding, informal triage could potentially accelerate the triage process, as such tools are quick and simple to use, but have never formally been tested for validity, reliability, and non-inferiority in comparison with—or as an addition to—formal triage tools. It was our hypothesis that informal triage using PDSR (by asking physicians “how ill does this patient look”) would perform similarly to formal triage using ESI, would have acceptable outcome validity and reliability, and could be used to increase predictive power if added to triage categories in case of “looking very ill”.

### 1.2. Goals of This Investigation

The primary objective of the present study was the direct comparison of PDSR at presentation to the ED with the well-established Emergency Severity Index (ESI), [12,13] using a non-inferiority testing of sensitivity and specificity regarding health-related outcomes (hospitalization, admission to an intensive care unit, and mortality). The secondary objectives were the reliability (inter-rater reliability between triage nurses and emergency physicians) and the predictive validity (outcomes according to five PDSR strata). Furthermore, safety (defined as mortality in patients with low risk attribution in either tool) and the improvement of the predictive validity of the ESI, if additionally stratified by the use of PDSR, were assessed. 

## 2. Materials and Methods

### 2.1. Study Design and Setting

Data were prospectively collected between 2013 and 2017 as part of the EMERGE cohort, which serves quality control purposes and benchmarking between Swiss EDs. All patients presenting to the ED of the University Hospital Basel, Switzerland 24 h a day, 7 days a week for three periods of three weeks were included by a dedicated and previously trained study team. This urban academic ED with a yearly census of over 50,000 patients is responsible for all emergencies, except for pediatric, ophthalmological, and obstetric patients, who are treated nearby. 

Team triage: A registered triage nurse was teamed with a board-certified emergency physician as previously described. For triage, the German version of the ESI algorithm was used. ESI levels were assigned following the four decision points, A to D, of the ESI triage algorithm: ESI level 1 (need of an immediate life-saving intervention (decision point A)), ESI level 2 (high-risk situation, new onset of confusion, lethargy, disorientation, or severe pain or distress (decision point B)), and ESI levels 3, 4, and 5 (more than one, one, or no resources needed (decision point C)). Before assigning ESI level 3, vital signs must be measured (decision point D). If they are outside of predefined limits (heart rate >100/min, respiratory rate >20/min, or oxygen saturation <92%), assigning an ESI level 2 must be considered. 

### 2.2. Physician Disease Severity Rating (PDSR)

For study purposes, there was an initial implicit judgement of disease severity before triage. There was no formal training for this implicit judgement. The raters had a written instruction to take only seconds for this judgement, and not to start with history taking or triage assessments before taking a number between 0 and 10 in order to answer the question “How ill does this patient look?”. A physician, unaware of the study goals, was present at the front door for this implicit judgement. This physician’s disease severity rating (PDSR) was used for the assessment of the primary goal, the comparison between PDSR and ESI. If both the triage nurse and physician were present, both individually rated the patient’s disease severity. These ratings were used for the secondary goal, inter-rater reliability.

### 2.3. Selection of Participants

All patients presenting consecutively to the ED during the study period were eligible after they gave their informed consent for inclusion. Patients who did not consent, who were directly referred to other departments, who left without being seen, who were screened twice, or who could not be screened were not included. Patients for whom the formal ESI triage level or the PDSR was not available were excluded from the analysis. Patients were included from October 21st to November 11th, 2013, February 1st to February 23rd, 2015, and January 30th to February 23rd, 2017, as these periods are representative for the case-mix over a year in the respective ED. The study protocol was approved by the local ethics committee (EKNZ-236/13) and complies with the Helsinki declaration.

### 2.4. Methods of Measurement

Upon arrival, all patients were registered by the study team, and an electronic health record (EHR) was opened. At presentation, the attending triage physician was first asked to rate how ill the patient appeared on a numeric scale ranging from 0 (completely healthy) to 10 (extremely ill), similar to the assessment of pain, without any further test results or clinical examination. This rating, the Physicians’ Disease Severity Rating (PDSR), was recorded on a printed machine-readable questionnaire without informing the physicians of its intent and purpose. The PDSR was converted to an ordinal five-level scale, inverse to triage scores. For reliability testing, triage nurses recorded disease severity ratings independently, but in exactly the same way as the physicians did, also before formal triage was started. Physicians and nurses worked as a team, and thus examined the patient in the same room but gave their ratings individually, unaware of each other’s ratings. Demographic information on age, gender, disposition, and in-hospital-mortality was retrieved from the EHR. Patients were followed for 30 days after their index presentation to the ED using information gathered from patients’ EHRs, the official municipal registry, insurance data, and written or telephonic contact with patients, proxies, or primary care providers.

In order to assess safety, we examined the reasons for system failures. System failure was defined as mortality in patients with low acuity (i.e., low PDSR (1 and 2) or low ESI levels (4 and 5)). All cases with low acuity who died within 30 days were analyzed. Two triage experts (having practiced ESI-Triage for many years and worked in the ER for more than 5 years) reviewed original triage nurses’ notes or handover protocols of direct boarders, and independently assigned an ESI level. Triage experts were blinded to the research question of the study and all other patient data, in addition to diagnoses and outcomes. If experts disagreed on an ESI assignment, consensus was reached with a referee by discussing the case.

In order to assess the relative change of predictive validity of PDSR (on top of ESI), we divided each ESI stratum into two groups (high PDSR vs. non-high PDSR groups for each respective ESI). High PDSR was defined as PDSR 5, the patients judged to look extremely ill. Non-high PDSR was defined as all other categories (PDSR 1–4).

In order to assess for the non-inferiority of PDSR vs. ESI, we compared the area under the receiver operating characteristic (ROC) curve (AUC) for paired ROC curves for each individual predefined outcome, as described below.

### 2.5. Outcome Measures

The primary outcome was the predictive validity of informal PDSR and formally assigned ESI. Predictive validity was defined as the ability of these tools to predict predefined outcomes—hospital admission, ICU admission, and mortality—as surrogates of “true” acuity [14].

Hospitalization was defined as direct transfer from the ED to any hospital ward. Any admission to an ICU, an intermediate care unit, or a stroke unit was defined as ICU admission. In-hospital mortality was defined as death at any time during hospitalization. Thirty-day mortality was defined as death within thirty days after presentation to the ED.

### 2.6. Primary Data Analysis 

All statistical analyses were carried out using the R software environment for statistical computing (*R* version 3.4.4; https://www.R-project.org/). PDSR was converted to an ordinal five-level scale—ranging from level 1, “appearing healthy”, to level 5, “appearing extremely ill”—in order to allow for direct comparison with the five-level ESI. Receiver operating characteristic (ROC) curves were plotted with ESI and PDSR as binary classifiers for all predefined outcomes. 

### 2.7. Noninferiority Method

In order to show the non-inferiority of PDSR as compared to ESI, a (two-sided) 95% confidence interval of the AUC difference between the corresponding ROC curves was calculated. Details are described in Pepe’s paradigmatic statistical textbook (Pepe, M.S. (2003) *The statistical evaluation of medical tests for classification and prediction*. Oxford: Oxford University Press).

The noninferiority level was set to −0.01. Hence, noninferiority can be concluded if the lower limit of the confidence interval is >−0.01.

Where applicable, Mann–Whitney U testing was used to compare the equality of distribution, and Fisher’s exact test of independence was used for comparison of proportions. Interrater-reliability between physician and nurse disease severity ratings was assessed by calculation of an intra-class correlation coefficient (ICC) based on a single-type, consistency, one-way random effects model. *p*-values < 0.05 were considered significant. ICCs of ≥0.6 were considered to represent good reliability, and ≥0.8 very good reliability [15]. 

### 2.8. Secondary Data Analysis

The secondary outcomes were the reliability, the predictive validity, and the safety of PDSR. For reliability, we used intraclass correlations between the disease severity ratings by physicians versus those of nurses. For predictive validity, we assessed all predefined outcomes stratified by PDSR. Furthermore, we analyzed associations between PDSR and ESI using cross-tabulation and by calculating percentages of overestimation and underestimation of risk using ESI as the “golden standard”. Overestimation of risk was given as the percentage of patients receiving a lower ESI category as compared to the PDSR category, divided by the total number of patients. Underestimation of risk was given as the percentage of patients receiving a higher ESI category as compared to the PDSR category, divided by the total number of patients. Perfect agreement was defined as the percentage of the patients that scored identically in both systems (the “diagonal” in the cross-tabulation; PDSR 3 = ESI 3). Relative agreement was defined as a one-step disagreement (PDSR 3 = ESI 2,3, or 4). The safety analysis was defined as an exploratory analysis of the percentages of mortality in low-risk PDSR and low-risk ESI patients. 

## 3. Results

### 3.1. Characteristics of Study Subjects

Out of the 8564 patients presenting to the ED, 7401 patients were screened for inclusion. We excluded 270 patients due to missing consent, double enrollment, or missing or incorrect data. Out of the 7131 patients enrolled, we excluded 14 patients who were not undergoing formal ESI-triage and 258 patients that were not rated by a physician, therefore without PDSR. The final analysis included 6859 patients, of which 355 (5.2%) were lost during the 30 day follow-up (see Figure 1).

The median age was 51 years (interquartile range (IQR) = 33 to 72 years); 51.4% were males. There were 159 non-survivors (2.4%) at the 30 day follow-up, of which 105 patients (1.5%) died in-hospital. Table 1 provides information on the distribution of ESI and PDSR levels and their respective hospitalization, ICU admission, and mortality rates.

### 3.2. Main Results

PDSR and ESI showed a comparable discriminatory ability [16] regarding the outcomes, as shown in Figure 2. All lower confidence interval (CI) limits were >−0.01, and noninferiority was concluded for all four endpoints: For hospitalization, the estimate of the AUC difference was 0.012 (CI 0.00046 to 0.024); for ICU admission, the estimate of the AUC difference was 0.016 (CI −0.0045 to 0.037); for in-hospital mortality, the estimate of the AUC difference was 0.034 (CI −0.00046 to 0.068); for 30 day mortality, the estimate of the AUC difference was 0.056 (CI 0.027 to 0.084). 

Interrater-reliability of PDSR (physicians vs. nurses on an individual patient level) showed an ICC of 0.61 (95% confidence interval (CI): 0.59–0.63).

Assessment of safety: Among the 2456 (35.8%) patients with low-acuity ESI (4 and 5), four (0.2%) died, and among the 4254 (62.0%) patients with low-acuity PDSR (1 and 2), seventeen (0.4%) died within 30 days of presentation. All four patients with a low ESI level who died had a low PDSR, while among the other thirteen low-PDSR patients who died, one had an ESI of 1, two an ESI of 2, and ten had an ESI of 3. The assessment of system failure showed that experts endorsed ESI 2 in two out of four cases of ESI 4 with mortality, as shown in Table 2. 

A comparison of ESI and PDSR on an individual patient level, demonstrated in Table 3, showed that 2459 of 6859 (36%) assessments showed perfect agreement, and 5750 of 6859 (84%) assessments showed relative agreement of +/– one step. An asymmetric distribution was found, with a tendency towards lower risk prediction using PDSR. Taking ESI as the “golden standard”, 53% of physician disease severity ratings would have underestimated the risk, and 11% overestimated the risk.

Assessment of the relative change of predictive ability: For each category in ESI 1–3, the comparison between high PDSR and non-high PDSR showed significant differences in both in-hospital und 30 day mortality, the highest differences being 36/86 (43.9%) for 30 day mortality in ESI 1/PDSR 5 versus 9/79 (12.7%) in ESI 1/PDSR <5 (*p* < 0.01), and 20/129 (16.1%) in ESI 2/PDSR 5 versus 31/1349 (2.4%) in ESI 2/PDSR <5 (*p* < 0.01). Comparisons were not possible in ESI 4–5 due to the low mortality (see Table 4).

## 4. Limitations

This study has several limitations: It is a single-center study. Therefore, external validity is limited. Furthermore, team triage or triage-liaison physicians are not ubiquitously available. In this study, nurses and physicians rated disease severity individually before formal triage, but in the same location. Therefore, it cannot be guaranteed that the physicians’ disease severity ratings were not influenced by the nurse’s severity ratings and vice versa. However, they rated independently, and we could not find obvious consequences of them being influenced by one another’s ratings. Third, informal triage by PDSR and subsequent formal triage by ESI were performed by the same team. Therefore, a high-acuity PDSR could well have influenced the ESI—maybe it should have influenced the assignment of ESI levels, considering the results shown. On the other hand, a “high-acuity” triage could have influenced the PDSR. However, we carefully planned the study to follow the same procedure in all patients: First, disease severity was judged independently by nurses and physicians, and, subsequently, formal triage was performed and documented on another page of the study questionnaire. Finally, physicians and nurses were not aware of the study aims, namely, to compare outcome validity between PDSR and ESI, and the PDSR did not lead to any procedural consequences, such as “immediate treatment” or “may wait”. However, it cannot be excluded that a high rating led to an immediate treatment of that patient if deemed necessary by the physician in charge. The PDSR was recorded on a Numeric Scale similar to that of a pain scale in order to render scoring quick and simple and to disguise the purpose of the study. However, some triage staff could have guessed that PDSR was to be compared to ESI. To our knowledge, no major ED has ever replaced formal triage with informal triage, as formal triage using a valid and reliable tool is the gold standard. Any other method to be used has to be compared to this standard. For comparison between the ESI and PDSR, we chose a non-inferiority approach, as it was not the primary goal to replace formal triage, but to investigate alternative tools. Such tools may support triage decisions under certain circumstances, such as crowding or shortage in nursing staff. Therefore, this was not a comparison of outcomes of formal versus informal triage, but merely a hypothesis-generating approach.

## 5. Discussion

The main finding of this study is the non-inferiority of PDSR compared to ESI triage regarding predictive validity in an all-comer ED population. However, both systems have weaknesses: In our cohort, partially due to system failures of ESI, mortality in “low-acuity” ESI 4 and 5 patients (0.2% in real life, 0.1% according to expert triage) was not completely excluded. Furthermore, the predominance of the ESI 3 category (40.2%) and the slim discrimination regarding mortality between the ESI 2 and ESI 3 categories (3.6 vs. 2.2%) are problematic. The weakness of PDSR was shown by the failure to exclude mortality in “low-acuity” PDSR 1 and 2 patients (0.4%) and the only moderate to good reliability of PDSR.

Due to the fact that about half of all patients are triaged as moderate acuity [17,18,19,20] and that upgrading the ESI is suggested in the case of “clinical suspicion” [21], we assessed the ESI’s predictive validity with and without information gained by PDSR (ESI sub-stratified by addition of PDSR 5 (“patient looks extremely ill”), see Table 4). Surprisingly, substantial differences were found in all ESI categories with relevant mortality. Taken together, the addition of high PDSR to any given ESI stratum increased the predicted mortality by a factor of 3.5 to 7. The recommendation to consider up-triage only in ESI 3 patients with clinical suspicion of serious outcomes [22] seems to fall short of the calamitous outcomes shown in all ESI strata with high-acuity severity ratings (“looking extremely ill”/PDSR 5). Furthermore, without a specific tool, upgrades may be difficult to implement in clinical practice [23].

Discrimination rates of under 1% regarding mortality between ESI 2 and ESI 3 [24] make it difficult to argue that patients in the lower-acuity category may safely wait for work-up, while patients in the higher-acuity category require almost immediate attention. Furthermore, the failure to exclude 30 day all-cause mortality in “low-acuity” ESI 4 and 5 patients has not yet been reported [13]. Of note, all system failures—using ESI or PDSR—occurred in older patients. For ESI triage, this is a well-documented phenomenon: Undertriage remains a problem [23], and even specifically designed programs do not seem to produce much benefit [25]. First, high-risk situations tend to be overlooked in older patients. Additionally, vital sign interpretation tends to be inadequate. Furthermore, disease presentation in older patients is often nonspecific, which is most likely the main reason for why high-risk situations are not recognized. In Table 3, some typical “atypical presentations” can be observed; most strikingly, patients presenting with weakness, gait instability, or falls ultimately suffer from severe disease or trauma. 

For PDSR, system failures in older patients have not been described, but may be explained by nonspecific disease presentation (weakness, dizziness, gait instability) [26], underestimation of geriatric syndromes (urinary retention, falls), and seemingly isolated issues (epistaxis, urinary catheter insertions). 

In summary, PDSR is a valid and reliable tool with acceptable safety. The non-inferiority of the discriminatory ability (PDSR versus ESI) regarding hospitalization, intensive care, and mortality could be shown. However, this is a proof-of-concept study, and both the feasibility and performance need to be tested in prospective trials, head-to-head against the gold standard. PDSR may be an additional tool to be used in times of crowding, since “waiting for triage” is a safety issue, but formal triage cannot be replaced on the basis of this novel evidence.

## Figures and Tables

**Figure 1 jcm-09-00762-f001:**
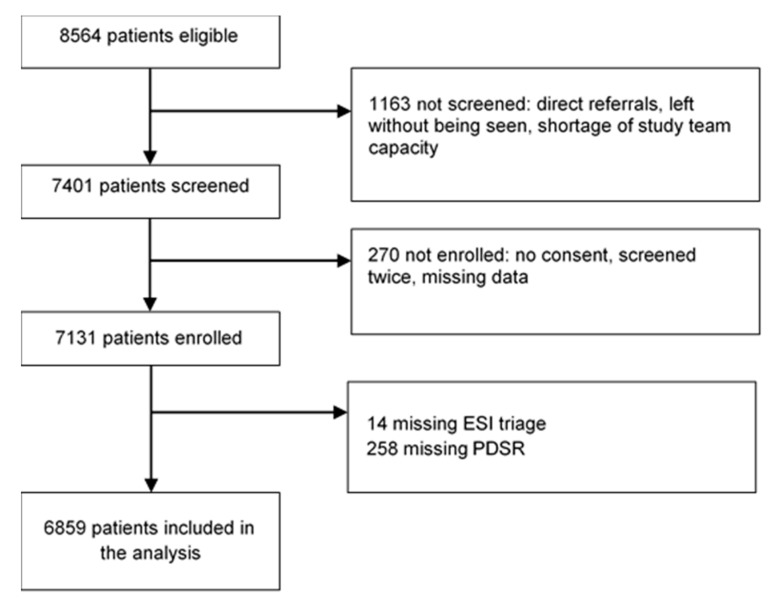
Inclusion procedure. ESI = Emergency Severity Index; PDSR = Physicians’ Disease Severity Rating.

**Figure 2 jcm-09-00762-f002:**
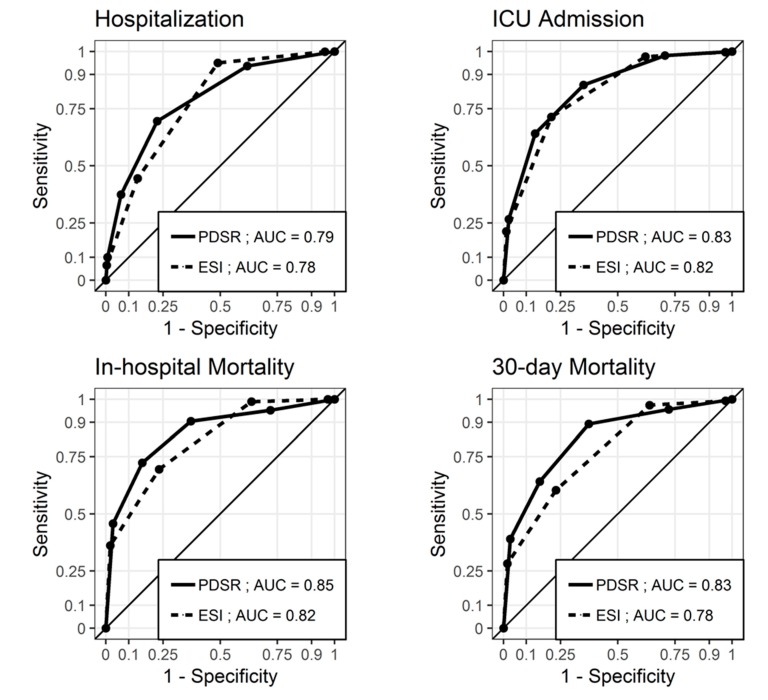
Receiver operating characteristic (ROC) curves for ESI and PDSR regarding hospitalization, ICU admission, in-hospital mortality, and 30 day mortality. The ESI level was coded inversely for the receiver operating characteristic curves to lie above the diagonal. PDSR = Physicians’ Disease Severity Rating; ESI = Emergency Severity Index; AUC = Area under the ROC curve; ICU = Intensive care unit.

**Table 1 jcm-09-00762-t001:** Baseline characteristics of the study population, stratified by Emergency Severity Index and Physicians’ Disease Severity Ratings, and respective outcomes (outcome validity).

	N (%)	Age, Median (IQR)	Male %	Hospitali- zation (%)	ICU Admission *n* (%)	In-hospital Mortality *n* (%)	30 Day Mortality, *n* (%) *
**All patients**	6859	51 (33–72)	51.4	2265 (33.0)	409 (6.0)	105 (1.5)	159 (2.4)
**ESI**							
1	165 (2.4)	68 (55–80)	58.8	148 (89.7)	87 (52.7)	38 (23.0)	45 (29.4)
2	1478 (21.5)	62 (44–77)	53.9	859 (58.1)	205 (13.9)	35 (2.4)	51 (3.6)
3	2760 (40.2)	58 (39–77)	49.1	1146 (41.5)	108 (3.9)	31 (1.1)	59 (2.2)
4	2260 (33.0)	38 (27–53)	51.9	109 (4.8)	8 (0.3)	1 (0.1)	3 (0.1)
5	196 (2.9)	40 (27–56)	52	3 (1.5)	1 (0.5)	0 (0.0)	1 (0.5)
**PDSR**							
5	260 (3.8)	70 (52–82)	56.5	226 (86.9)	109 (41.9)	48 (18.5)	62 (24.8)
4	891 (13.0)	66 (48–80)	53.5	622 (69.8)	153 (17.2)	28 (3.1)	40 (4.7)
3	1454 (21.2)	60 (41–77)	47.9	727 (50.0)	87 (6.0)	19 (1.3)	40 (2.8)
2	2354 (34.3)	49 (32–68)	51.4	546 (23.2)	53 (2.2)	5 (0.2)	10 (0.4)
1	1900 (27.7)	40 (28–57)	52.2	144 (7.6)	7 (0.4)	5 (0.3)	7 (0.4)

Baseline characteristics of the study population, stratified by the Emergency Severity Index and Physicians’ Disease Severity Ratings. ESI = Emergency Severity Index; PDSR = Physicians’ Disease Severity Rating (from 1, not ill, to 5, extremely ill); IQR = Interquartile range; ICU = Intensive care unit. * Lost to follow-up patients are shown as survivors.

**Table 2 jcm-09-00762-t002:** Formal and informal triage categories in non-survivors with low acuity.

Age	Gender	PDSR	ESI	Rater A	Rater B	Reference ESI	Reason for Presentation	Cause of Death
74	Male	2	1	2	2	2	Treated hypoglycemia	Bronchial carcinoma
87	Male	2	2	2	2	2	Weakness, Anemia	Leukemia, fungal pneumonia
89	Male	2	2	2	2	2	Aspiration	*Unknown*
90	Female	2	3	2	2	2	Weakness	Influenza A
87	Female	2	3	2	2	2	Syncope	Tachy-/bradycardia syndrome, severe aortic valve stenosis
70	Male	2	3	2	2	2	Dizziness	Septic shock, bilateral pneumonia
97	Female	2	3	2	3	2	Leg pain, Dyspnea	Septic shock, heart failure
77	Male	2	3	3	4	3	Back pain	Heart failure
79	Female	2	3	3	3	3	Epistaxis	Kidney abscess
69	Female	1	3	3	3	3	Epistaxis	Suicide
90	Male	1	3	2	2	2	Low energy fall	Myocardial infarction, pneumonia
89	Male	1	3	3	3	3	Aspiration	Heart failure, pneumonia
91	Female	1	3	2	2	2	Gait instability	Bronchial carcinoma, delirium
86	Female	2	4	2	3	2	Low energy fall	Subdural hematoma
68	Male	1	4	4	4	4	Urinary catheter insertion	*Unknown*
72	Male	1	4	3	4	4	Discomfort by urinary catheter	*Unknown*
86	Male	1	5	2	4	2	Urinary catheter insertion	Delirium

Thirty-day non-survivors with low acuity. Low acuity was defined as a low Emergency Severity Index (4 and 5) or a low Physicians’ Disease Severity Rating (1 and 2). Reference ESI represents the consensual Emergency Severity Index level assigned by two independent experts blinded to the real-life ESI, as well as to any other outcome. In case of disagreement, a consensual ESI level was obtained by discussing the case. ESI = Emergency Severity Index; PDSR = Physicians’ Disease Severity Rating.

**Table 3 jcm-09-00762-t003:** Cross-tabulation of PDSR and ESI.

	ESI				
PDSR	5	4	3	2	1
**1**	141	1105	512	137	5
**2**	48	907	1051	338	10
**3**	5	204	841	390	14
**4**	2	41	314	484	50
**5**	0	3	42	129	86

Direct comparison between PDSR and ESI. ESI 5 is the lowest acuity and risk stratum, corresponding to PDSR 1. ESI 1 is the highest acuity and risk stratum, corresponding to PDSR 5.

**Table 4 jcm-09-00762-t004:** Patient characteristics and risk stratification according to ESI. ESI strata were substratified by the addition of PDSR 5 (“patient looks extremely ill”).

			N	Age, Median (IQR)	Male %	Hospitalization, *n* (%)	ICU Admission, *n* (%)	In-hospital Mortality, *n* (%)	30 day Mortality, *n* (%)*
**ESI 1**	**PDSR**	**=5**	86	71 (58–80)	61.6	84 (97.7)	55 (64.0)	31 (36.1)	36 (43.9)
		**<5**	79	68 (54–80)	55.7	64 (81.0)	32 (40.5)	7 (8.9)	9 (12.7)
**ESI 2**	**PDSR**	**=5**	129	70 (50–82)	56.6	104 (80.6)	45 (34.9)	14 (10.9)	20 (16.1)
		**<5**	1349	61 (43–77)	53.6	755 (56.0)	160 (11.9)	21 (1.6)	31 (2.4)
**ESI 3**	**PDSR**	**=5**	42	73 (52–83)	47.6	36 (85.7)	8 (19.0)	3 (7.1)	6 (14.6)
		**<5**	2718	58 (39–77)	49.1	1110 (40.8)	100 (3.7)	28 (1.0)	53 (2.0)
**ESI 4**	**PDSR**	**=5**	3	57 (48–62)	33.3	2 (66.6)	1 (33.3)	0 (0)	0 (0)
		**<5**	2257	38 (27–53)	52.0	107 (4.8)	7 (0.3)	1 (0.1)	3 (0.1)
**ESI 5**	**PDSR**	**=5**	0	-	-	-	-	-	-
		**<5**	196	40 (27–56)	52.0	3 (1.5)	1 (0.5)	0 (0.0)	1 (0.5)

Characteristics of patients with the highest PDSR versus all others in each ESI category. The table shows age, gender, and adverse outcomes. ESI = Emergency Severity Index; PDSR = Physicians’ Disease Severity Rating; IQR = Interquartile Range; ICU = Intensive care unit. * Lost for follow-up patients are shown as survivors.

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
