# Peer review of "Physicians’ Disease Severity Ratings are Non-Inferior to the Emergency Severity Index"

_jcm, 2020, doi:10.3390/jcm9030762_

Round 1
Reviewer 1 Report
Would be interested to see how much time this adds to the triage process. Also would be interested to see how people with less training would perform. One of the limitations could be the lack of generalization to providers with less experience. Would like more information regarding the agreement between physician and nurse and see if the discrepancy is more pronounced in certain subpopulations, specifically the elderly. Given that most death were in older adults it would be interested in more discussion regarding their under triage. Not all patients had full vital signs in table 2. Is this standard at your hospital to not have documented RR, temp, etc on all patients?
Author Response
Thank you for your all your questions and inputs. We will answer them point by point below:
Would be interested to see how much time this adds to the triage process.
Answer: Physician disease severity ratings add very little time to the triage process. It can be compared to a numeric rating of pain (0-10), only that patients do not need to be interviewed. The "thin slicing" theory claims that only split seconds are necessary for such implicit judgments. In reality, it could take a little longer, as the number needs to be recorded in the EHR. We have not recorded the exact time used for PDSR, but according to experience, after thousands of cases, an average duration of 10 seconds can be assumed.
Also would be interested to see how people with less training would perform.
Answer: None of the physicians and nurses were specifically trained for this Task. We have added this Information to the methods section. We have used this method before a few years ago without Trainings and have compared nurses and senior physicians rating the same patients (Int J Clin Pract. 2015 Jun;69(6):710-7). The area under the curve of the receiver operating characteristic curves was 0.77 for experienced physicians and 0.72 for nurses (hospitalisation), and 0.72 for exp. physicians and 0.68 for nurses (acute morbidity). Though physician ratings showed significantly better outcome validity, we do not believe that the difference is relevant in clinical practise. Pilot data showed that Junior physicians had comparable outcome validity to senior physicians (PDSR strata regarding hospitalization, ICU admission, in-hospital, and 30day mortality).
In the present study, PDSR were all made by physicians. Unfortunately, we can not provide any direct comparisons between junior and senior physicians in this sample, as decisions had to be made quickly at the door and during clinical routine.
One of the limitations could be the lack of generalization to providers with less experience.
Answer: As pointed out above, senior physicians and nurses have similar Performance using disease severity ratings, if outcomes were considered. As there is no real immediate feedback mechanism, the "thin slicing" method tends not to improve with more general medical experience.
Would like more information regarding the agreement between physician and nurse and see if the discrepancy is more pronounced in certain subpopulations, specifically the elderly.
Answer: In an all-comer population, the agreement between physician an nurse is described above. Intra-class correlation was 0.49 which is moderate. We have not studied the discrepancy between nurses and physicians in the elderly, but in a cohort with 1278 patients with nonspecific complaints and a median age of 82 years, PDSR had a validity of .67 (discriminatory rate .91) for mortality and .68 (DR .91) for acute morbidity, which is comparable to formal triage. If the reviewer wishes, we could provide a reliability analysis in patients 65 and over, e.g. in an online-appendix.
Given that most death were in older adults it would be interested in more discussion regarding their under triage.
Answer: We have previously studied undertriage of older emergency patients https://journals.plos.org/plosone/article?id=10.1371/journal.pone.0106203 ; https://www.sciencedirect.com/science/article/pii/S0196064411019287?via%3Dihub and found that there are 2 important aspects that lead to under triage. First, high-risk situations were not recognized. Additionally, vital sign interpretation was inadequate. Disease presentation in older patients is often nonspecific https://doi.org/10.1111/j.1553-2712.2009.00658.x which is most likely the main reason why high risk situations are not recognized. In table 3, some typical “atypical presentations” can be observed, most strikingly patients presenting with weakness, gait instability or falls ultimately suffering from severe disease or trauma.
We have added a paragraph discussing the reasons for undertriage in older adults.
Not all patients had full vital signs in table 2. Is this standard at your hospital to not have documented RR, temp, etc on all patients?
Answer: As we are using the ESI for triage we only document vital signs in higher acuity patients, strictly adhering to the tool.
Reviewer 2 Report
Dear Dr. R. Bingisser,
Manuscript ID jcm-718686 entitled “Physicians’ Disease Severity Ratings are Non-inferior to the Emergency Severity Index” has been reviewed. The authors report the comparing triage performance between Physicians’ Disease Severity Ratings (PHDS) and Emergency Severity Index (ESI) at the Emergency department. While the findings presented in this study should be of interest to this journal audience, it is my opinion that a major revision, based on the technical comments given below, is needed to make this manuscript suitable for publication.
Major points:
- In this study, you said to compare the accuracy of discriminatory ability between PDSR and ESI as primary analysis, and to compare the reliability as secondary analysis. However, it is difficult to understand how the data were analyzed in detail. Moreover, in the methods section, you showed only the primary data analysis. Please show what the objective was and the type of examination method you used to prove this concretely.
- In this manuscript, you have used the ESI algorithm. However, I cannot clearly understand the PDSR algorithm, for example, “a person” (physician or nurse?) and the timing when physicians or nurses performed the PDSR method are unclear. Please show the PDSR algorithm using a follow-chart or figure.
- Please state the reason why you have used the different scale-level: a 10-scale-level at the time of triage and a 5-scale-level at the time of analysis. Moreover, please indicate the associations among the triage scale-level, analysis scale-level, and ESI levels using a table.
- I cannot understand why you compared two groups (high PDSR vs. non-high PDSR group) to assess the relative change in predictive validity between PDSR level 5 and ESI level 1. Please indicate the validity of the examination method by consulting with statistical specialist.
- I cannot understand why you have compared two groups (survivors vs. non-survivors) in the results. If you show the comparison in the results, please describe the purpose of this comparison and the statistical analysis methods used in methods section.
- Please define the terminology used in this manuscript, for example, “top of ESI” and “first impression”.
- I cannot understand the reason why both the physician and nurse performed the PDSR assessment in the same patients and why two triage experts assigned the ESI level, which was assessed by physician or nurse.
- In the discussion session, please discuss about the primary and secondary outcome mainly.
Author Response
Thank you for your valuable input. We have considered each point together with our professional biostatistician and believe that we can answer each point in order to improve the submission.
- In this study, you said to compare the accuracy of discriminatory ability between PDSR and ESI as primary analysis, and to compare the reliability as secondary analysis. However, it is difficult to understand how the data were analyzed in detail. Moreover, in the methods section, you showed only the primary data analysis. Please show what the objective was and the type of examination method you used to prove this concretely.
Answer: Thank you for the chance to improve the manuscript.
We have added the hypotheses (lines 49-52) and we have more clearly described the objectives of the study (lines 50-56) and the methods used to check the hypotheses. Of note, this is a hypothesis generating study, as it is the first of its kind. We wanted to explore, if risk stratification could be made using a brief implicit judgement ("how ill does this patient look"), using a non-inferiority approach. Further, we have tested, if the addition of a high PDSR could improve the predictive power of the ESI. We have extended the methods section by a more detailed description of the study design and setting (lines 75-79), of the primary analysis using non-inferiority testing (lines 132-137), as well as the secondary analysis (inter-rater reliability between nurses and physicians) (lines 139-143). The third assessment was a safety analysis, defined as an exploratory analysis of mortality in low-risk PDSR and ESI patients (lines 143-144). We have further added a table comparing PDSR and ESI on an individual patient level (new table 2). - In this manuscript, you have used the ESI algorithm. However, I cannot clearly understand the PDSR algorithm, for example, “a person” (physician or nurse?) and the timing when physicians or nurses performed the PDSR method are unclear. Please show the PDSR algorithm using a follow-chart or figure.
Answer: We routinely use the ESI algorithm for over 10 years in our institution. Therefore, in this study, ESI was performed as routine task by trained triage nurses. Additionally, as a research question, PDSR, an implicit judgement, taking only seconds to perform, was done by a physician, unaware of the study goals. PDSR can not be called an algorithm, as it only answers the question "how ill does this patient look" on a numeric rating scale of 0 to 10. This has now been described in more detail instead of referring to previous data of our group (lines 79-89). In a subset of over 4000 patients, we have asked the triage nurses, additionally to emergency physicians at the front door, to take down disease severity ratings for study reasons. They were also not informed about the study goals. These data points were only used for reliability testing. - Please state the reason why you have used the different scale-level: a 10-scale-level at the time of triage and a 5-scale-level at the time of analysis. Moreover, please indicate the associations among the triage scale-level, analysis scale-level, and ESI levels using a table.
Answer: First, as we have previously used PDSR for other cohorts and shown its usability, we have continued to use a 10-point numeric rating scale for this study, as it resembles the NRS for pain. Second, it seemed too obvious to ask for disease severity ratings on a 5-point scale and immediately after use a 5-point triage scale. We did not want to take the risk of confounding the two scales and the risk of a "spill-over" effect. However, for reasons of comparison to the 5-point-scale ESI, it was necessary to collapse the 10-point PDSR to 5 points, because a 10 point scale tends to perform better than a 5-point scale due to more data points, if comparing receiver operating curves according to our statistician. Therefore, we chose to test for non-inferiority of PDSR (collapsed to 5 points, and inverse to ESI) with the original 5-point ESI. We have replaced table 2 by a new cross-tabulation of PDSR and ESI, in order to indicate the associations on an individual patient level. We have calculated over- and underestimation of risk using PDSR as compared to ESI (as "golden standard"), as described in the methods. - I cannot understand why you compared two groups (high PDSR vs. non-high PDSR group) to assess the relative change in predictive validity between PDSR level 5 and ESI level 1. Please indicate the validity of the examination method by consulting with statistical specialist.
Answer: Thank you for this important question that we have also discussed with our biostatistician. PDSR high was defined as 9 or 10 points on the numeric rating scale anwering the question "how ill does this patient look". Using the collapsed PDSR, 5 points were attributed to these exytremely ill-looking patients. According to our hypothesis, predictive power could be increased by splitting all ESI levels (the "extremely ill looking" PDSR 5 vs. all other patients of the respective triage category). We have added a sentence (lines 275 to 278) to the discussion to better explain this approach.
The problem of formal triage is the low discriminatory ability between the two most frequent categories, namely ESI 2 and ESI 3. Multiple attempts have been made to improve predictive validity of ESI (e.g. adding ADL, coherence of history or first impression, see West J Emerg Med. 2019 Jul;20(4):633-640), the most promising being first impression.
Taken together, not only ESI 1 category was split into "Very ill looking" PDSR 5 and all other ESI 1, but all other ESI categories as well.
Predictive validity increased by 3.5 to 7 times if adding high PDSR to ESI in all categories with mortality >1% which supported the hypothesis that informal triage using PDSR may be added to formal triage for better risk stratification. - I cannot understand why you have compared two groups (survivors vs. non-survivors) in the results. If you show the comparison in the results, please describe the purpose of this comparison and the statistical analysis methods used in methods section.
Answer: We agree with the reviewer that the comparison of the two groups (survivors vs. non-survivors) does not add much to the manuscript. Except for the age, no relevant differences were found. We therefore deleted table 2 and replaced it by the cross-tabulation of PDSR vs. ESI. We have mentioned in the results section that the difference between survivors and non-survivors was mainly the age. More Information could be provided in an online appendix, if the reviewer would favour this. - Please define the terminology used in this manuscript, for example, “top of ESI” and “first impression”.
Answer: We would like to excuse for the misunderstanding. The sentence on line 61 reads "...the relative change of predictive ability of PDSR on top of the ESI...". There is no "top of the ESI". In order to reduce the possibility of misunderstanding, we have re-written the sentence.
"First impression" was replaced by "disease severity rating", which is defined in the methods section. - I cannot understand the reason why both the physician and nurse performed the PDSR assessment in the same patients and why two triage experts assigned the ESI level, which was assessed by physician or nurse.
Answer: The subset of over 4000 patients that was rated by both physicians and nurses was used to test reliability. It was virtually impossible to have a second physician attend at the door in a study of 9 weeks around the clock. Therefore, inter-rater reliability was tested between physicians and nurses, knowing that there is a difference and that outcomes can be predicted significantly better by physicians (see: Int J Clin Pract. 2015 Jun;69(6):710-7).
The second question regarding the re-assignment of ESI levels by triage experts is a research question, possibly interesting to researchers in the field of triage. It is still debated why older patients tend to suffer from under-triage. According to the safety assessment using an exploratory analysis of the mortality of "low-risk" patients, we hereby present the first data on mortality in "low-risk" ESI patients. Therefore, the question needed to be resolved, if low risk patients were under-triaged due to incorrectly applying the ESI algorithm, or due to a System failure of the ESI algorithm. The re-assessment of the triage notes of all fatalities was performed by triage experts in order to classify the failures of both systems. While ESI showed acceptable safety in the low-risk categories, two cases, with short term mortlity after ED discharge, aged 68 and 72, could be identified that were formally correctly triaged as ESI 4. We believe that this table is worth showing for better understanding of the risk analysis. - In the discussion session, please discuss about the primary and secondary outcome mainly.
Answer: We realize that we have extended the discussion to the validity of ESI triage. However, we believe that these points should not be omitted, as the data presented on 30 day outcomes after ESI triage are among the larger outcome studies in this field. Issues, such as the slim discrimination regarding mortality between the ESI 2 and ESI 3 categories (3.6 vs. 2.2%) are of importance to many practising emergency physicians, not only researchers. Further, the daily discussions on how to improve ESI triage, particularly the question on how to identify ESI 3 patients at risk of serious outcome, could benefit from the data presented.
In order to shorten the discussion, we have removed two paragraphs on outcomes and future research. On the other hand, we have added a paragraph on undertriage in older patients due to the request of reviewer 1.
We would like to thank reviewer 2 for extremely helpful input. We believe that this manuscript has considerably improved after revisions.
Round 2
Reviewer 2 Report
Dear Dr. R. Bingisser,
Thank you for the opportunity to review this manuscript.
Manuscript ID jcm-718686 entitled “Physicians’ Disease Severity Ratings are Non-inferior to the Emergency Severity Index” has been re-reviewed.
This second version of the manuscript has been much improved in accordance to my comments, and is in a nice condition. I recommend that it be accepted for publication.
I wish you continued success in the future.